# Evaluation of the Attractant Effect of *Solanum muricatum* (Solanales: Solanaceae) on Gravid Female Adults of *Zeugodacus tau* (Diptera: Tephritidae) and Screening of Attractant Volatiles

**DOI:** 10.3390/insects14070591

**Published:** 2023-06-30

**Authors:** Pingfan Jia, Xiaoyu Zhang, Bo Wang, Qinge Ji

**Affiliations:** 1Biological Control Research Institute, Plant Protection College, Fujian Agriculture and Forestry University, Fuzhou 350002, China; 2200203007@fafu.edu.cn (P.J.); xiaoyuzhang1022@163.com (X.Z.); 2China Fruit Fly Research and Control Center of FAO/IAEA, Fuzhou 350002, China; 3Key Lab of Biopesticide and Chemical Biology, Ministry of Education, Fuzhou 350002, China; 4State Key Laboratory of Ecological Pest Control for Fujian and Taiwan Crops, Fuzhou 350002, China

**Keywords:** *Zeugodacus tau*, *Solanum muricatum*, induced oviposition, attraction rate, volatile

## Abstract

**Simple Summary:**

The Solanaceae crops are the main hosts of *Zeugodacus tau*, a pest insect that causes serious damage to economically important crops of Solanaceae. In the winter–spring period, female adults of *Z. tau* have been found to oviposit on the pepino melon *Solanum muricatum*. However, the differences in the ability of the fruits of *S. muricatum* and other Solanaceae crops to attract gravid *Z. tau* females have not been clarified. In this study, oviposition and trapping bioassays were performed to clarify whether such differences existed. GC-MS was used to identify the compounds and relative contents of the volatile odors of *S. muricatum*. The Y-tube olfactometer system was used to determine the compounds that induced behavioral responses in gravid *Z. tau* females. The results show that *Solanum muricatum* odor was attractive for *Z. tau* females. The nine compounds from the volatiles of *S. muricatum* induce behavioral responses in gravid *Z. tau* females. These results are beneficial for preventing *Z. tau* females from harming *S. muricatum*, developing attractants and repellents for *Z. tau* females, and establishing a system of ecological control for *Z. tau* females.

**Abstract:**

*Zeugodacus tau* (Walker) (Diptera: Tephritidae) is a pest seriously harmful to Solanaceae crops and was found to oviposit on the pepino melon *Solanum muricatum* (Aiton). To date, the differences in the ability of the fruits of *S. muricatum* and other Solanaceae crops to attract gravid *Z. tau* females have seldom been reported. Oviposition and trapping bioassays were performed to clarify whether such differences existed. A combination of GC-MS and the Y-tube olfactometer system was used to identify and determine the compounds inducing behavioral responses in gravid *Z. tau* females to the volatile odors of *S. muricatum*. The results show that *S. muricatum* odors play a role in attracting gravid *Z. tau* females. The odors of Solanaceae crops influence their ability to attract these organisms. The nine compounds from the volatiles of *S. muricatum* induce tendency or repellency responses in gravid *Z. tau* females. Hexyl acetate, butyl acetate, amyl actate, and isoamyl acetate caused tendency behavior in gravid *Z. tau* females, while hexyl hexanoate, butyl isovalerate, butyl valerate, and isoamyl hexanoate caused repellency behavior. Heptyl acetate caused repellency behavior in gravid *Z. tau* females at higher concentrations (5 mg/mL) but caused tendency behavior at a low concentration (0.5 mg/mL). These results suggest that vigilance against the harm caused by *Z. tau* is required during the cultivation of *S. muricatum*. The nine compounds of the volatile odors of *S. muricatum* could help to develop attractants and repellents for gravid *Z. tau* females. These results are beneficial for preventing *Z. tau* females from harming *S. muricatum*, developing attractants and repellents for *Z. tau* females, and establishing a system of ecological control for *Z. tau* females.

## 1. Introduction

*Zeugodacus tau* (Walker) (Diptera: Tephritidae), formerly known as *Bactrocera tau*, is a polyphagous pest of fresh fruits and vegetables and has been listed as a quarantine pest in many countries, including Japan, the Republic of Korea, Indonesia, Pakistan, and Jordan [1,2]. *Zeugodacus tau* was first recorded in China in Fujian province in 1849 by Walker [1,2]; now, it is an economically important agricultural pest and mainly infests fruits of Cucurbitaceae, Moraceae, Myrtaceae, Sapotaceae, and Solanaceae in tropical and sub-tropical Asia and the South Pacific region [3,4,5]. The female adults of *Z. tau* achieve explosive population growth by laying a large number of eggs; they can lay roughly 9.9 eggs at once, on average, and 464 eggs over a lifetime [6]. The newly hatched larvae of *Z. tau* burrow and feed inside the flesh, making the firm and tender flesh loose and porous, resulting in the decay and shedding of the immature fruit, making the fruit inedible and economically useless [7]. The quality and yield of vegetables and fruits are severely reduced in areas where *Z.tau* is distributed. *Zeugodacus tau* has caused 21–34% and 21–32% infestations of *Siraitia grosvenorii* (Cucurbitales: Cucurbitaceae) and *Cucurbita moschata* (Cucurbitales: Cucurbitaceae), respectively, in Taiwan and China [8], and approximately 5.6% yield losses of ripened *Luffa acutangula* (Cucurbitales: Cucurbitaceae) in Thailand have been caused by *Z. tau* [9]. High survival rates for *Z. tau* at four different life stages from egg to adult, were shown to be maintained under extreme cold and heat, with temperatures spanning −5 to 0 °C and 39 to 42 °C, respectively [2,7]. With the warming of the climate [10], the areas suitable for *Z. tau* establishment are gradually increasing, increasing the risk of its invasion and spread. In recent years, *Z. tau* has become increasingly harmful to Solanaceae crops; in 2014 and 2015, the annual losses of tomato *Solanum lycopersicum* (Solanales: Solanaceae) plants caused by *Z. tau* in India were approximately 13.72–59.77% and 15.86–69.89%, respectively, and the degree of damage showed an increasing trend year by year [11].

The pepino melon *Solanum muricatum* (Aiton) (Solanales: Solanaceae) is a perennial herbaceous horticultural species and a close relative of the tomato, native to the Andean region, and has been cultivated artificially since the Pre-Columbian era [12,13]. The ripened fruits have a light flavor, with a fragrant and sweet taste, making them suitable for consumption in desserts and as an ingredient of salads, juices, or ice cream [14,15]. The fruit is low in calories, low in starch and sugars, high in phenolics, and rich in minerals (Fe, Zn, Cu, Mn, Ca, and P) [16]. It has long been recognized as an edible and medicinal fruit owing to its antioxidant, anti-tumor, antidiabetic, anti-inflammatory, and hypotensive properties [14,15]. For the past several decades, pepino melon had not been officially recognized as a host for *Z. tau* [1]; this could be because *Z.tau* was not distributed in the Andes of South America, a major area of pepino melon cultivation [12,13]. In recent years, the area used for *S. muricatum* cultivation in China has increased by nearly 200,000 hectares due to the crop’s high edible and medicinal value [17,18,19]. The extensive cultivation area for *S. muricatum* overlaps with the distribution area of *Z. tau* [1,19]. In the winter–spring period, female adults of *Z. tau* were found to oviposit on the pepino melon in Fuzhou City of China (Appendix A). It is necessary to determine what underlies the potential of *S. muricatum* to attract *Z. tau* females to prevent the damage caused by *Z. tau* because *Z. tau* has a strong tendency and capacity to harm other crops in the Solanaceae family [1,11]. In previous studies, primary processed products (such as juice and fruit peel fragments) of more attractive host fruits had stronger attractant effects than the control when used for trapping female fruit flies [20,21,22,23]. If the Solanaceae crops with the optimal induced-oviposition effects are also effective in trapping, this will further contribute to targeting the factors that influence the induced-oviposition attraction of the pests to Solanaceae crops and help us to use the primary processed products of Solanaceae crops for the field trapping of the *Z. tau* females.

Meanwhile, screening candidate compounds of female attractants from the volatiles of host plants is considered to be a particularly promising and feasible approach due to the odors containing the key kairomone enabling females to recognize host plants [24,25,26,27,28,29,30,31,32,33,34,35]. To date, the behavioral responses of economically important fruit fly (Diptera: Tephritidae) pests, such as *Rhagoletis pomonella* [24,25], *Ceratitis capitata* [26,27], *Bactrocera dorsalis* [28,29,30], *B. correcta* [30], *B. trynoi* [31], *Zeugodacus cucurbitae* [20,32], *B. minax* [33], *Anastrepha ludens* [34], and *A. obliqua* [35], to volatile compounds of host fruit odors have been studied using GC-MS (gas chromatography–mass spectroscopy) in combination with behavioral bioassays or electrophysiological bioassays. In addition, several groups of female attractants made from a mixture of volatile odor compounds have achieved trapping effects in fields [20,25,36]. In a field trial, the captures of female *Z. cucurbitae* with an attractant consisting of nine compounds from the volatiles of cucumber were twice those achieved using traps baited with Solulys protein bait [20]. Subsequently, a new attractant consisting of seven compounds from the volatiles of cucumber captured more adults of *Z. cucurbitae* than traps baited with Solulys protein bait [36]. The field captures of female *R. pomonella* in traps baited with an attractant consisting of five compounds from the volatiles of apple were better than those achieved with butyl hexanoate [25]. Raspberry ketone (RK) and zingerone (ZN), screened from *Bulbophyllum* (Asparagales: Orchidaceae) plants, have been widely applied as male attractants in trapping *Z. tau* males [37]. There are few reports on the volatile compounds of host fruit odors that attract gravid *Z. tau* females to their hosts [1]. If several compounds that cause behavioral reactions in gravid *Z. tau* females in response to Solanaceae fruit odors are identified, this could contribute to the development of attractants for gravid *Z. tau* females.

Therefore, in this study, the ability of *S. muricatum* and other Solanaceae crops to attract the females of *Z. tau* was comprehensively evaluated via induced oviposition and trapping experiments. GC-MS was used to determine the chemical constituents of the volatile odors of *S. muricatum* fruit. The chemical compounds that cause behavioral responses in gravid *Z. tau* females were selected using Y-tube olfactometer bioassays. The results of this study provide a basic data reference for preventing *Z. tau* females from harming *S. muricatum*, developing attractants for *Z. tau* females, and establishing a system of ecological control for *Z. tau* females.

## 2. Materials and Methods

### 2.1. Insect Colonies

The *Z. tau* was collected from a mixed fruit–vegetable plantation of Zhuqi in Minhou City, Fujian Province, China, and maintained at the UN (China) Center for Fruit Fly Prevention and Treatment, Fujian Agriculture and Forestry University. The female flies of *Z. tau* have an externally sclerotized ovipositor at the end of the abdomen, which enables them to be distinguished from males [1]. *Zeugodacus tau* was permitted to oviposit in a soft plastic bottle containing 100 mL of pumpkin juice that was neatly pierced with holes; the eggs were collected and transferred to a tray containing a mill feed diet for larval development [38,39]. The pumpkin juice, obtained from chopped pumpkin fruit (including pulp and skin) with water at 1 g:1 mL ratio, was liquidized in a blender in the laboratory. After allowing the flies to oviposit for 8 h, the eggs were collected from the bottle into rearing trays containing an artificial diet consisting of wheat, sugar, and yeast for larval development, prepared according to Chang et al. [38]. Then, the trays were placed into larger containers that contained 3 cm layers of moist, fine sand, in which the larvae pupated. The pupae were separated by sieving the sand and transferred into a screened cage (30 cm × 30 cm × 30 cm) until emergence. The fruit fly adults were provided with yeast extract, sugar (1:3, wt/wt), and moistened cotton swabs [39]. The yeast extract was purchased from Jiehui Chemical Co., Ltd. (Jinan, China). The insectary was maintained at 26 ± 1 °C with 65 ± 5% relative humidity (RH) and a 12:12 h (L:D) photoperiod.

### 2.2. Solanaceae Crops

The Solanaceae crops such as tomato *S. lycopersicum*, eggplant *S. melongena*, pepper *Capsicum annuum*, pepino melon *S. muricatum,* and its different cultivars were purchased from Fuzhou Yonghui supermarket. Fruits with intact skin without damage were selected for the experiments, soaked in clean water for 2 h, rinsed three times using running water, and dried to remove surface moisture. The trade names of the tomato *S. lycopersicum* cultivars used in the experiments included fleshy tomato (FT), mushy tomato (MT), and cherry tomato (CT). The trade names of the eggplant *S. melongena* cultivars used in the experiments included violet-black long eggplant (VBLE), violet-black round eggplant (VBRE), ordinary long eggplant (OLE), and linear eggplant (LE). The trade names of the pepper *C. annuum* cultivars used in the experiments included screw pepper (SP), linear pepper (LP), bell pepper (BP), and long green pepper (LGP). The trade names of the pepino melon *S. muricatum* cultivars used in the experiments included pepino melon (PM).

### 2.3. Chemicals

The 20 species experiment compounds that could be bought (Table 1) were purchased from Aladdin Industrial Corporation (Shanghai, China) or Macklin Biochemical Technology Co., Ltd. (Shanghai, China) and had a purity of 95.0% or higher (with the exception of cis-6-nonenyl acetate (>92.0%)).

### 2.4. Oviposition Bioassays

The induced-oviposition effects of pepino melon and other Solanaceae crops on gravid *Z. tau* females were evaluated using oviposition bioassays [40]. The mean number of eggs oviposited by *Z. tau* females in other hosts is 9.9 eggs/female; this was taken as a reference to evaluate the induced-oviposition effects [6]. Three treatment groups (Treatment A1: four cultivars of pepper; Treatment A2: three cultivars of tomato; Treatment A3: four cultivars of eggplant) were used to screen for the optimal cultivars regarding the number of eggs laid for the three species of Solanaceae crops. Thirty pairs of adults (25 d old) of *Z. tau* were captured in acrylic transparent cages (30 cm × 30 cm × 30 cm). The peel (30 g), excluding the seeds and placenta, was wrapped in plastic wrap in which oviposition notches (2 cm × 2 cm) were cut on both sides, respectively. The prepared peels were placed in disposable plastic Petri dishes (d = 9 cm) and arranged in a transparent acrylic cage at equal intervals for 8 h (9:00–17:00). In the first three treatments, the tomato, pepper, and eggplant cultivars with the optimal numbers of eggs laid were selected, respectively, and were used for experiments together with pepino melon as Treatments A4-A6. In Treatment A4, the peels of 4 species of Solanaceae crops were prepared in the same way as for Treatments A1-A3. In Treatment A5, compared with Treatment A4, the intact fruit, excluding the calyx and pedicel, was wrapped in plastic wrap in which an oviposition notch (2 cm × 2 cm) was cut on the calyx side. In Treatment A6, compared with treatment A4, the peel fragments (30 g) were placed in an egg-laying bottle that was made from a cylindrical soft plastic bottle (d = 3.8 cm, h = 12.0 cm, vol. = 100 mL), neatly pierced with holes; this was used to induce the females of *Z. tau* to oviposit (Figure 1A). The number of eggs laid by the females of *Z. tau* for each species or variety was counted under a stereomicroscope, and the number of single female eggs was calculated. The treatments were repeated six times.

### 2.5. Trapping Bioassays

The trapping effects of the odors of pepino melon with other Solanaceae crops on gravid *Z. tau* females were evaluated using trapping bioassays [40]. Gravid (25–35 d old) females (*n* = 45) of *Z. tau* were captured in acrylic transparent cages (30 cm × 30 cm × 30 cm). The peel fragments (10 g), excluding the placenta and seeds, were placed in a special trap combined with a white, opaque plastic bottle (d = 5.7 cm, h = 12.8 cm, vol. = 250 mL) with a lid and three centrifuge tubes (d = 1.0 cm, h = 4.0 cm, vol. = 2 mL) (Figure 1B). The trap had one hole in the center of the cap and holes in the upper and lower parts of the bottle so that the centrifuge tube with the bottom third removed could be inserted as an entry channel for the females. The holes in the upper and lower parts of the bottle were positioned opposite each other and were 3 cm away from the top and bottom of the bottle, respectively. Traps that contain different peel fragments were arranged in the acrylic transparent cage at equal intervals for 8 h (9:00–17:00) and removed and frozen for 2 h. The number of female *Z. tau* in each trap was counted, and the attraction rates were calculated (attraction rates % = the number of attracted/the total number of experiments). The treatments were repeated six times per peel fragment. Three treatment groups (Treatment B1: four cultivars of pepper; Treatment B2: three cultivars of tomato; Treatment B3: four cultivars of eggplant) were used to screen out the optimal cultivars regarding the attraction rates for the three species of Solanaceae crops. In the first three treatments, the tomato, pepper, and eggplant cultivars with the optimal attractant rates were selected, respectively, and were used for experiments together with pepino melon as Treatment B4.

### 2.6. Identification of Volatile Compounds from the Fruit of S. muricatum

Headspace volatile samples of pepino melon fruits were obtained and quantified using gas chromatography–mass spectroscopy [12]. The peel fragments (5 g) were placed in a headspace sample bottle (vol. = 40 mL) closed with a silicone rubber mat following a 10 min incubation period at 40 °C in a water bath. The volatile compounds were collected using SPME (65 µm polydimethylsiloxane-divinylbenzene coated fiber) at room temperature for 50 min. Three biological replicates were performed per sample. The volatiles were separated and identified using a GC-MS system, the Shimadzu GCMS-QP2020NX instrument coupled to a 0.25 µm Rtx-5MS fused silica capillary column with a 30 m × 0.25 mm inner diameter. Helium (1.1 mL/min) was used as the carrier gas. The injector temperature was 250 °C, and it was set for split (2:1) injection. The temperature program for the headspace volatile samples was as follows: holding at 40 °C for 5 min, increasing by 3 °C/min to 50 °C, holding for 3 min, increasing by 5 °C/min to 150 °C, increasing by 15 °C/min to 250 °C, and holding for 5 min. Helium was used as the carrier gas at a flow rate of 1 mL/min. Mass spectra for 35–550 *m*/*z* were obtained at 70 eV with an ion source at 230 °C.

The compounds of the volatiles of pepino melon were tentatively identified by the comparison of the mass spectrum with the authenticated reference standards and with spectra in the National Institute for Standard and Technology (NIST 2017) mass spectral library and in the Mass Spectra of Flavors and Fragrances of Natural and Synthetic Compounds (FFNSC), and the relative contents of compounds were calculated using the area normalization method. The retention indices (RI), which were calculated by injecting an n-Alkanes mixture (C7-C40) into the GC-MS for each compound, were compared with the RI values on the website (https://webbook.nist.gov/chemistry/name-ser/ (accessed on 17 December 2022), https://www.flavornet.org/flavornet.html (accessed on 17 December 2022), or http://www.odour.org.uk (accessed on 17 December 2022)) [41,42,43]. The mean values were calculated via triplicate analysis [41].

### 2.7. Olfactometer Bioassays

A Y-tube olfactometer system [44,45,46,47,48,49] was used to investigate the behavioral reactions of gravid *Z. tau* females to the volatile compounds from the fruit of *S. muricatum*. The Y-tube olfactometer system consisted of air pump that produced a unidirectional airflow; activated charcoal bottles that filtered air; and distilled water bottles that humidified air, flowmeters that controlled the speed of airflow at 200 mL/min, volatile source bottles, a Y-shaped glass tube (length of main stem: 20 cm; length of each arm: 20 cm; branching angle: 60°; inner diameter: 3 cm), and silicone tubing that connected all the components of the system. Volatile source bottles connected to the arm of the Y-shaped glass tube were placed inside a white polypropylene plastic hollow plate cage (length = 90 cm; width = 35 cm; height = 40 cm) to eliminate environmental interference with the behavior of the gravid *Z. tau* females. A piece of sterile gauze (2 cm × 2 cm) was used in the junction between the volatile source bottle and the rubber hose to prevent the females from entering the rubber hose. A wide-spectrum fluorescent lamp (8 W, 45 cm long) was positioned outside the plate cage and directly above the volatile source bottle. For 30 min prior to the experiments, air was left flowing through the activated charcoal and distilled water bottles to achieve equilibrium in the two arms and the Y-tube stem [44]. Then, control tests using pure air in two arms were carried out to confirm that the Y-tube olfactometer was assembled correctly [45]. The bioassays were performed between 09:00 and 17:00 h in a darkened room, and the temperature and humidity were adjusted to 26 ± 1 °C and 65 ± 10%, respectively. A releaser chamber consisting of two centrifugal tubes (vol. = 10 mL and vol. = 50 mL) was used to pretreat and release each group of gravid *Z. tau* females. Groups of 20 gravid *Z. tau* females in the release chamber were introduced to the assay room for 30 min to acclimatize them prior to each experiment [46,47]. Each compound was diluted and mixed with reagents with four concentration gradients (5 mg/mL, 0.5 mg/mL, 0.05 mg/mL, and 0.005 mg/mL; all of the compounds used were dissolved in paraffin liquid), using paraffin liquid as the control. The reagent and solvent were released by placing 200 uL onto filter papers (2 cm × 2 cm) in the volatile source bottles, respectively. Five minutes after the release chamber had been attached to the olfactometer, gravid *Z. tau* females were released into the stem of the Y-tube and allowed 30 min to choose between the control and reagents. The gravid *Z. tau* females that moved 3 cm toward the stem of the Y-tube were regarded as responders; when they entered an arm tube, moved at least 3 cm, and remained there for 30 s, this was regarded as an effective choice [44]. The means of three recordings for the numbers of responders and effective choices were taken every 10 min over 30 min, recording the numbers of responses and choices of the gravid *Z. tau* females in each group. In each case, the percentage of insects that made a distinct choice was calculated (responses rate % = the number of responders/the total number of females; choice rate % = the number of effective choices/the total number of females; tendency rate % = the number of effective choices to treatment; repellency rate % = the number of effective choices to control). Each group of females in the experiment was used only once, and the Y-tube and volatile source bottle were cleaned with absolute ethanol and dried in a 70 °C oven for 20 min after each experiment [48]. After one group had been observed, the Y-tubes and the volatile source bottle were replaced with clean tubes, and the Y-tube and reagent were rotated 180° to minimize any positional bias effects [48]. Each concentration of reagent was tasted four times [49].

### 2.8. Data Analysis

The experimental data were analyzed using the package IBM SPSS Statistics version 23 (SPSS, Inc., Chicago, IL, USA). The data of the oviposition bioassays and trapping bioassays were analyzed after checking that the data were normally distributed and that there was homogeneity of variances [39]. Percentage data were arcsine square-root-transformed for further statistical analysis; untransformed data are presented in tables and figures [39]. The number of single female eggs laid and the attraction rates were analyzed via one-way ANOVA. Differences between crops were analyzed via pairwise comparisons using a one-way ANOVA with a Student–Newman–Keuls (S-N-K’s) test at *p* < 0.05. In the olfactometer bioassays, the data from the choice of gravid *Z. tau* females were first transformed into percentages and then analyzed using paired chi-square (χ^2^) tests (McNemar’s) [50]. A diagram of the bottle was drawn using Inkscape. A peak chromatographic plot was drawn using Origin version 8. Other figures were drawn using GraphPad Prism version 8 (GraphPad Software).

## 3. Results

### 3.1. Difference in Oviposition of the Gravid Z. tau Females Induced by S. muricatum with That Induced by Other Solanaceae Crops

The induced-oviposition effects on gravid *Z. tau* females were significantly different for different cultivars of peppers, tomatoes, and eggplants. The numbers of eggs laid by gravid female *Z. tau* was significantly different across four cultivars of eggplants, and the highest number of eggs laid (6.07 eggs/female) was on the VBRE cultivar (*F*= 59.527, *df* = 3, *p* = 0.000) (Figure 2A). The number of eggs laid by gravid female *Z. tau* was significantly different across three cultivars of tomatoes, and the highest number of eggs laid (32.68 eggs/female) was on the CT cultivar (*F* = 71,672.570, *df* = 2, *p* = 0.000) (Figure 2B). The number of eggs laid by gravid female *Z. tau* was significantly different across four cultivars of peppers, and the highest number of eggs laid (5.86 eggs/female) was on the LP cultivar (*F* = 1272.991, *df* = 3, *p* = 0.000) (Figure 2C). The induced-oviposition effects of eggplants and peppers were inferior, as the gravid *Z. tau* females oviposited fewer than 9.9 eggs/female for those.

The induced-oviposition effects on gravid *Z. tau* females were significantly different for the four species of Solanaceae crops. The numbers of eggs laid by gravid female *Z. tau* was significantly different across four species of Solanaceae crops, and the highest number of eggs laid (37.51 eggs/female, 37.15 eggs/female, and 3.61 eggs/female) were on peel fragment, complete fruit, and odor bottle of the pepino melon (*F* = 122,721.586, *df* = 3, *p* = 0.000; *F* = 919.813, *df* = 3, *p* = 0.000; and *F* = 358.821, *df* = 3, *p* = 0.000) (Figure 2D–F, respectively). The odor of pepino melon was a pivotal and potential cue for ensuring the optimum induced-oviposition effect for gravid *Z. tau* females.

### 3.2. Differences in Trapping of gravid Z. tau Females for S. muricatum and Others Solanaceae Crops

The trapping effects for gravid *Z. tau* females significantly differed for different cultivars of peppers, tomatoes, and eggplants. The rates at which gravid female *Z. tau* was attracted were significantly different across four cultivars of eggplants odor, and the highest rate (12.59%) was on the VBRE cultivar odor (*F* = 23.717, *df* = 2, *p* = 0.000) (Figure 3A). The rates at which gravid female *Z. tau* was attracted were significantly different across three cultivars of tomato odor, and the highest rate (54.08%) was on the CT cultivar odor (*F* = 23.717, *df* = 2, *p* = 0.000) (Figure 3B). The rates at which gravid female *Z. tau* was attracted were significantly different across four cultivars of peppers odor, and the highest rate (22.96%) was on the BP cultivar odor (*F* = 5.097, *df* = 3, *p* = 0.009) (Figure 3C).

The trapping effects on gravid *Z. tau* females significantly differed for four species of Solanaceae crops. The rates at which gravid female *Z. tau* was attracted were significantly different across four species of Solanaceae crops, and the highest (56.30%) was on the Pepino melon odor (*F* = 40.968, *df* = 3, *p* = 0.000) (Figure 3D).

### 3.3. Volatile Compounds from the Fruit of S. muricatum

A total of 47 volatile compounds from the pepino melon peel fragment were tentatively identified via gas chromatography–mass spectroscopy, which could be separated into 32 esters, 5 alcohols, 3 ketones, 2 alkanes, 1 alkylphenol, and 4 heterocycles (Table 2 and Figure 4). Of all these volatiles, esters were the most abundant compounds, with the highest value of 91.37%. A total of 12 volatile compounds with relative contents greater than 1%, including butyl acetate, isoamyl acetate, 3-methyl-3-buten-1-ol acetate, (Z)-2-penten-1-ol acetate, amyl acetate, hexyl acetate, 4-methylcyclohexanol acetate, heptyl acetate, cis-6-nonen-1-ol, nonyl alcohol, cis-6-nonenyl acetate, and nonyl acetate, were the main contributors to the odor of pepino melon. Of these, hexyl acetate was the most abundant compound, with a relative content of 52.27%. A total of 35 volatile compounds had relative contents less than 1%, and these may be complementary contributors to the signature odor of pepino melon.

### 3.4. Olfactory Behavior of the Gravid Z. tau Females in Response to Different Volatile Compounds

The 20 volatile compounds that were available and legal to purchase from chemical companies were tested in a Y-tube olfactometer system to evaluate the selected behavioral reactions of gravid *Z. tau* females. Nine of the twenty compounds tested in the Y-tube olfactometer were found to significantly elicit select behavioral responses in gravid *Z. tau* females (Figure 5 and Figure 6). Paraffin liquid solutions of hexyl acetate (5 mg/mL) (*p* = 0.007, χ^2^ = 7.507), butyl acetate (5 mg/mL; 0.5 mg/mL; 0.05 mg/mL) (*p* = 0.000, χ^2^ = 20.645; *p* = 0.000, χ^2^ = 28.502; *p* = 0.005, χ^2^ = 8.286), amyl acetate (5 mg/mL) (*p* = 0.006, χ^2^ = 8.069), and isoamyl acetate (0.05 mg/mL) (*p* = 0.029, χ^2^ = 4.846) were found to be significantly attractive to gravid *Z. tau* females (Figure 5A–C). Paraffin liquid solutions of butyl isovalerate (5 mg/mL; 0.5 mg/mL) (*p* = 0.019, χ^2^ = 5.860; *p* = 0.025, χ^2^ = 5.626), isoamyl hexanoate (5 mg/mL) (*p* = 0.017, χ^2^ = 6.020), hexyl hexamoate (5 mg/mL) (*p* = 0.009, χ^2^ = 7.474), and butyl valerate (5 mg/mL) (*p* = 0.025, χ^2^ = 5.760) were found to be significantly repellent to gravid *Z. tau* females (Figure 6A,B). The ninth compound was heptyl acetate, which was found to repel gravid *Z. tau* females at 5 mg/mL (*p* = 0.033, χ^2^ = 5.551) (Figure 5A) but attract them at 0.5 mg/mL (*p* = 0.019, χ^2^ = 6.196) (Figure 5B). The effect of isoamyl acetate on the behavioral responses of gravid *Z. tau* females was similar to that of heptyl acetate, but its repellent effect at high concentrations was not statistically significant (*p* = 0.791, χ^2^ = 0.246) (Figure 5A).

## 4. Discussion

Differences in the species and cultivars of Solanaceae crops had a significant impact on the aspect of induced oviposition and trapping of gravid *Z. tau* females. This is consistent with previous research on several polyphagous fruit flies (Diptera: Tephritidae), such as *B. zonata* [51,52], *Z*. *cucurbitae* [52,53], *A. obliqua* [54], *Neoceratitis cyanescens* [55], *B. oleae* [56], *B. tryoni* [57,58], *B. dorsalis* [23,40,59,60], and *B. latifrons* [61,62]; these studies found that the ability to attract fruit flies differed between species and cultivars of host plants. Polyphagous fruit flies with higher olfactory plasticity are able to recognize and use more hosts and choose their preferred host plants more than [63]. The previous studies indicated that hosts with greater attraction were advantageous to the offspring performance of fruit flies [51,53,56,57,58,60]. The previous studies verified a positive correlation between the attractiveness of Cucurbitaceae, Myrtaceae, and Rutaceae crops to *Z. tau* and the offspring performance (larval survival, pupation rate, and emergence rate) of *Z. tau* [64]. In this study, a series of experiments first evaluated the trapping and induced-oviposition effects of different species and cultivars of Solanaceae crops, the main host plants of gravid *Z. tau* females. Here, we showed that line pepper (LP), violet-black round eggplant (VBRE), and cherry tomato (CT) were the optimal cultivars regarding induced oviposition for gravid *Z. tau* females. Meanwhile, bell pepper (BP), violet-black round eggplant (VBRE), and cherry tomato (CT) were the optimal cultivars for trapping gravid *Z. tau* females. Furthermore, in contrast to the above optimal cultivars, our results showed that pepino melon showed optimal performance regarding induced oviposition and trapping for gravid *Z. tau* females. Whether there exists a positive correlation between the attractiveness of Solanaceae crops to *Z. tau* and the offspring performance of *Z. tau* requires further testing. In previous studies, primary processed products (such as juice and fruit peel fragments) of more attractive host fruits were used as attractants or baits to control female fruit flies [20,21,22,23]. Moreover, the attractant effect was better than that of the control when using primary processed products to trap female fruit flies [20,21,22,23]. In a previous study, guava was recommended for attracting *B. dorsalis* in papaya and banana gardens during the fruiting season [60]. Whether the primary processed products of *S. muricatum* fruit could be used as a primary attractant to trap gravid *Z. tau* females in the field needs to be further tested.

Fruit flies with an imminent oviposition-ready physiological status are more sensitive to and show stronger preferences for host fruit odors [40]. Therefore, it is easy for aggregation and egg oviposition to be induced by host plants with the optimal attractive ability [51]. In the experiments in which the optimal cultivars were grouped, the induced-oviposition and attractant rates regarding gravid *Z. tau* females for the optimal cultivars of pepper or eggplant were lower than 10 eggs/female and 30%, respectively, and for the optimal cultivar of tomato, these were more than 30 eggs/female and 50%, respectively. In the experiments in which the optimal species were grouped, the induced-oviposition and attractant rates regarding gravid *Z. tau* females for cherry tomatoes were second only to those for pepino melon but significantly superior to those for pepper and eggplant. It can be speculated that, in areas where Solanaceae crops without pepino melon are planted, the gravid *Z. tau* females may be more attracted to tomato’s odors and be more likely to lay eggs. Our results regarding the attractiveness of the host odors initially explain the reason why *Z. tau* has been the main pest of tomato crops in India. In the past several decades, pepino melon has not been recorded as a host for *Z. tau* [1]. This could be because *Z. tau* is not distributed in the Andes of South America, which is the main area in which pepino melon was cultivated [12,13]. In recent years, the areas in which *S. muricatum* is extensively cultivated have come to overlap with the area in which *Z. tau* is distributed [1,19]. Meanwhile, female adults of *Z. tau* were found to oviposit on the pepino melon in Fuzhou City of China during the winter–spring period (Appendix A). Therefore, those who manage the cultivation areas of pepino melon may need to be vigilant against the harm that *Z. tau* can cause during the cultivation period.

In this study, our results showed that the odor of Solanaceae crops is a key cue affecting their ability to attract gravid *Z. tau* females. Previous studies have shown that the color and odor of the host are two important physicochemical characteristics that influence the attraction of fruit flies [55,63,65]. The olfaction of fruit flies is an important contributor to host location in complex environments and over long ranges [63], while fruit flies’ vision was found to be more significant with regard to their host preferences over short ranges [63]. Visual and olfactory cues may interfere with the results obtained during experiments studying the host selection of fruit flies [55]. *Neoceratitis cyanescens* can sometimes show significantly different preferences between hosts that differ only in odor or color [55]. However, *N. cyanescens* can also sometimes show no particular preference between hosts that differ in two variables (color and odor) [55]. The different color preferences of *Z. tau* adults are significantly affected by chromatic cues. Virtual wavelengths of 568 nm (yellowish green) were shown to be the optimum wavelengths for trapping mature *Z. tau* [65]. It is necessary to control for host physical and chemical characteristics that may interfere with the results of experiments, such as color. In a previous study, a method that used a “jar within a jar” device with fans effectively avoided the interference of color and other physical and chemical characteristics in experiments evaluating the attractiveness of *B. zonata* and *Z. cucurbitae* caused by host odor [52]. In this study, we achieved a similar goal using an opaque compact odor spawning and trap bottle, respectively. Our results show that the induced oviposition and attractant rates obtained with odor accord with those obtained for the complete fruit and peel fragment. The unique odor of pepino melon is a major attractant of gravid *Z. tau* females, and it is worth further investigating the odor of chemical compounds.

The combination of GC-MS with behavioral bioassays or electrophysiological bioassays is a classic method used to identify the effects of plant volatile compounds on the behaviors of insects [24,25,26,27,28,29,30,31,32,33,34,35]. In this study, our results show that 47 volatiles were identified in the pepino melon peel fragment and provided many candidate compounds that cause behavioral responses in gravid *Z. tau* females. Our results show that there were 35 esters in the volatile odor of *S. muricatum*, in contrast to the results of previous studies that only determined five esters [12,66]. In this study, the volatile odor came from cultivars that had yellow ripe round fruits in a large planted area of China but not from cultivars with fruit that had significantly different morphological characteristics and that had not been introduced for cultivation [12,66]. A previous study showed that the “Coche” mango cultivar, which showed an optimal ability to attract *A. obliqua,* has three more esters that are attractive to females than the “Ataulfo” mango cultivar [35]. Further experiments are needed to determine whether different cultivars or geographical environments affect the composition and relative contents of the volatile odor compounds of *S. muricatum*.

Host plants conducive to larval survival and development are selected by gravid female fruit flies for oviposition because the larvae are limited in their ability to spread [29]. Specific chemical cues in the host odor are accurately identified by gravid female fruit flies, which use their olfactory senses to locate and search for suitable hosts [35]. Individual or mixed compounds in the host odor that are considered to be specific chemical cues for attracting females are often screened as female attractants [20,32,33,35]. A synthetic blend combined with volatile odor compounds from two cultivars of mango was highly attractive for *B. dorsalis* females [35]. An attractant that consisted of nine compounds from the volatile odor of cucumber was the most attractive and resulted in the most captures of *Z. cucurbitae* females in baited field traps [20]. Similarly, an attractant that was mixed with volatile odor compounds from the ridge gourd was attractive to *Z. cucurbitae* females [32]. Four compounds from the citrus volatiles were identified via GC-MS (gas chromatography–mass spectroscopy), and using the Y-tube olfactometer system proved that they could significantly attract *B. minax* [33]. At present, there have been few studies on the chemical compounds that cause behavioral responses in gravid *Z. tau* females to their hosts. In this study, our results showed that nine compounds in the volatile odor of pepino melon could cause an olfactory behavioral response in gravid *Z. tau* females. The four compounds (hexyl acetate, butyl acetate, amyl actate, and isoamyl acetate) with relatively high contents were more likely to cause tendency behavior in gravid *Z. tau* females, while the four compounds (hexyl hexanoate, butyl isovalerate, butyl valerate, and isoamyl hexanoate) with relatively low contents were more likely to cause repellency behavior. High concentrations (5 mg/mL) of heptyl acetate with relative contents greater than 1% caused repellency behavior in gravid *Z. tau* females, while low concentrations (0.5 mg/mL) of heptyl acetate caused tendency behavior. Hexyl acetate was reported to affect the behavior or EAG responses of *B. dorsalis* [28], *A. ludens* [67], and *R. pomonella* [24], while isoamyl acetate and butyl acetate were also reported to affect the behavior and EAD responses of *B. dorsalis* [28]. It was first reported that amyl actate induced behavioral responses in fruit fly females. Interestingly, the total relative contents (3.61%) of heptyl acetate and the four compounds that caused repellency behavior in gravid *Z. tau* females were much lower than those (59.54%) of the four compounds that only caused tendency behavior. Meanwhile, a previous study showed that blends of compounds could be attracted towards the black bean aphid *Aphis fabae* (Hemiptera: Aphididae), although when compounds are tested individually, the compounds may actually act as repellents [68]. Therefore, it is clear that the natural proportions characteristics of various compounds in the volatile odor of pepino melon may make it an excellent attractant for gravid *Z. tau* females. This is also consistent with previous research showing that there may be multiple compounds, rather than a single compound, that make the host’s volatile odor an attractant [20,32,33,35]. A previous study showed that electrophysiological bioassays (GC-EAG) can help to locate specific volatiles in a mixture that the insect’s antennae are responding to, which can help to pinpoint which compounds specifically may be causing biological activity [32]. And it contributes to the screening of the attractant formulas that blend the attractive compounds that could be tested for attraction toward fruit flies [20,32,33,35]. Therefore, more efforts are needed to develop female attractants or repellents that are highly effective for gravid *Z. tau* females based on the volatile odor composition of pepino melon. The differences in the tendencies of hosts to attract fruit flies are affected by the species and relative contents of volatile compounds [20,32,33,35]. More systematic studies that combine GC-MS with principal component analysis are needed to determine whether differences in the effectiveness of Solanaceae crops in attracting gravid *Z. tau* females are related to these factors [42].

Pepino melon has the potential to act as a trap plant to attract and accumulate *Z. tau* females during the winter–spring period. The strong attraction of target insects to plants is a fundamental and basic condition when selecting trap plants [69,70,71]. Based on this, a perimeter trap plant barrier for hot cherry peppers was applied to protect a centrally located main crop of bell peppers from oviposition and infestation by the pepper maggot, *Zonosemata electa* (Diptera: Tephritidae) [72]. Furthermore, a trap to protect papaya groves, which are susceptible to infestation, was designed to reduce the damage caused by the papaya fruit fly *Toxotrypana curcicauda* (Diptera: Tephritidae) [73]. Similarly, intercropping squashmelon *Cucurbita melopepo* (Cucurbitales: Cucurbitaceae) with melon *Cucumis melo* (Cucurbitales: Cucurbitaceae) was performed as a trap to reduce the infestation of melons by the melon fly *Z. cucurbitae* [74]. The relative content of compounds in the volatile odor of pepino melon, which is an excellent attractant for gravid *Z. tau* females, is close to 60%, consistent with the fundamental and basic conditions for trap plants. In the winter–spring period, with the host being absent, the number of scattered *Z. tau* determined their schedule and the scale of the outbreak of the current year [75]. Attracting and aggregating *Z. tau* females in this period is a valuable but difficult task. In Fuzhou City of China, pepino melon can continue to grow, flower, and bear fruit in winter and spring. Meanwhile, *Z. tau* females were found to oviposit on pepino melon in Fuzhou City of China during the winter–spring period (Appendix A). Therefore, it is worth examining whether pepino melon can better attract *Z. tau* as a trap plant in the field. New research suggests that fruit flies can not successfully colonize passion flowers *Passiflora edulis* (Malpighiales: Passifloraceae) because their eggs fail to hatch, owing to the influence of the toxic compound hydrogen cyanide produced as part of the immune response of passion flowers [76]. In the summer–autumn period, the reproductive success rate of *Z. tau* females should be greatly reduced by the large-scale planting of passion flowers [76]. Therefore, this could be a new and effective ecological control method that uses different trap plants and their functions in different periods in which the population dynamics of *Z. tau* differ. It is hypothesized that increasing plant genetic diversity benefits pest control [77]. Therefore, discovering more trap plants with different ecological control functions with respect to *Z. tau* may be a meaningful future research direction.

## 5. Conclusions

Differences in the species and cultivars of Solanaceae crops had a significant impact on the induced oviposition and trapping of gravid *Z. tau* females. Pepino melon showed optimum performance regarding induced oviposition and trapping for gravid *Z. tau* females. Vigilance against the harm that *Z. tau* can cause during the cultivation of *S. muricatum* is important.

Our results show that the odor of Solanaceae crops is a key cue affecting their ability to attract gravid *Z. tau* females. The nine compounds in the volatile odor of pepino melon could cause olfactory behavioral responses in gravid *Z. tau* females. The natural proportions of various compounds in the volatile odor of pepino melon make it an excellent attractant for gravid *Z. tau* females.

In this study, we first systematically identified nine compounds from the host crop of *Z. tau* that could induce behavioral responses in gravid *Z. tau* females. The compounds with relatively high contents were more likely to cause tendency behavior in gravid *Z. tau* females, while the compounds with relatively low contents were more likely to cause repellency behavior. The nine compounds from the volatile odor of *S. muricatum* that induced behavioral responses could be used to develop attractants and repellents for gravid *Z. tau* females.

There is the potential for pepino melon to be used as a trap plant to attract and accumulate *Z. tau* females during the winter–spring period. In subsequent field applications, it is expected that pepino melon and passion flowers will be organized to establish a new and effective model for the ecological control of *Z. tau*.

As stated previously, the findings of this study provide basic data for vigilance regarding the harm that *Z. tau* females can cause in the cultivation of *S. muricatum*. The finding provides candidate compounds for the development of attractants and repellents for *Z. tau* females. At the same time, we preliminarily discussed how the excellent attraction properties of *S. muricatum* could be applied in the ecological control of *Z. tau* females.

## Figures and Tables

**Figure 1 insects-14-00591-f001:**
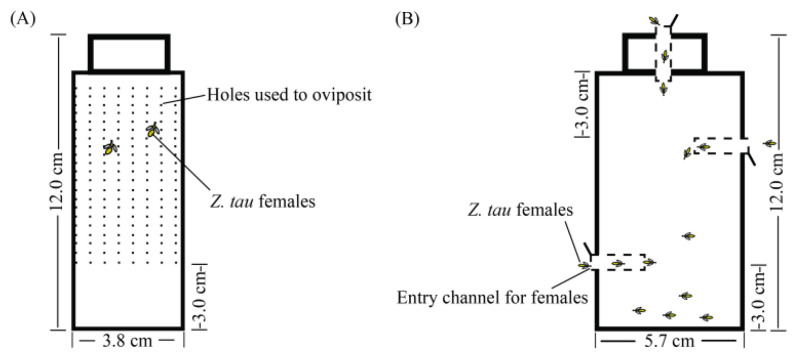
Diagram of the egg-laying bottle and the trapping bottle. (**A**) The egg-laying bottle. (**B**) The trapping bottle.

**Figure 2 insects-14-00591-f002:**
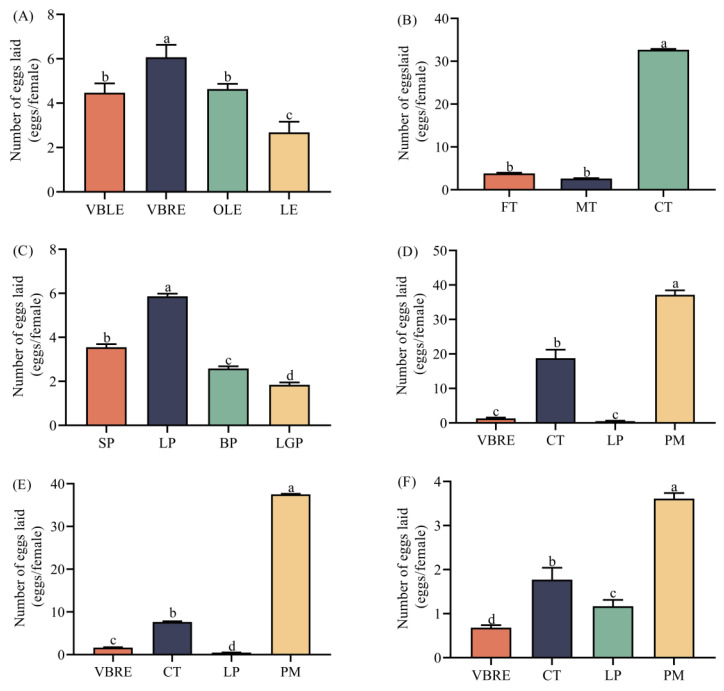
Numbers of eggs laid by the gravid *Z. tau* females with different Solanaceae crops. Different lowercase letters mean significant differences among different cultivars or species at the same time point, at *p* < 0.05. (**A**) Numbers of eggs laid by the gravid *Z. tau* females for different eggplant cultivars. (**B**) Numbers of eggs laid by the gravid *Z. tau* females for different tomato cultivars. (**C**) Numbers of eggs laid by the gravid *Z. tau* females for different pepper cultivars. (**D**) Numbers of eggs laid by the gravid *Z. tau* females for the complete fruit of different Solanaceae species. (**E**) Numbers of eggs laid by the gravid *Z. tau* females for the peel fragments of different Solanaceae species. (**F**) Numbers of eggs laid by the gravid *Z. tau* females for the fruit odors of different Solanaceae species. FT: Fleshy tomato; MT: Mushy tomato; CT: Cherry tomato; VBLE: Violet-black long eggplant; VBRE: Violet-black round eggplant; OLE: Ordinary long eggplant; LE: Linear eggplant; SP: Screw pepper; LP: Linear pepper; BP: Bell pepper; LGP: Long green pepper; PM: Pepino melon.

**Figure 3 insects-14-00591-f003:**
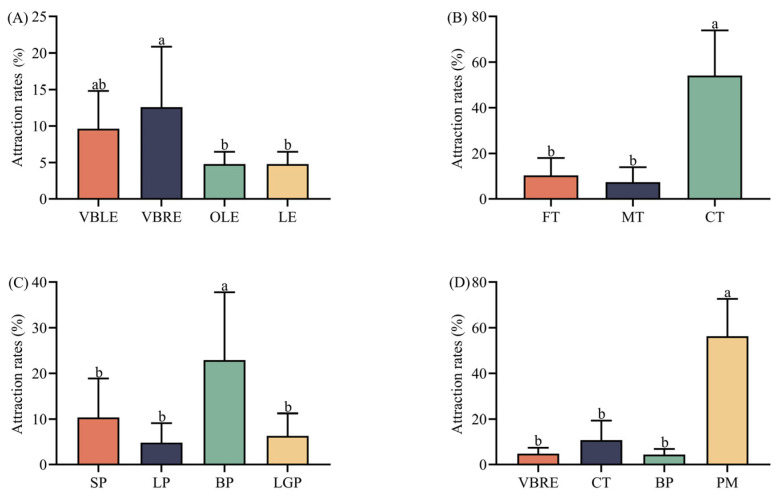
Attraction rates for the gravid *Z. tau* females for different Solanaceae crops. Different lowercase letters mean significant differences among different cultivars or species at the same time point, at *p* < 0.05. (**A**) Attraction rates for the gravid *Z. tau* females for different eggplant cultivars. (**B**) Attraction rates for the gravid *Z. tau* females for different tomato cultivars. (**C**) Attraction rates for the gravid *Z. tau* females for different pepper cultivars. (**D**) Attraction rates for the gravid *Z. tau* females for the fruit odors of different Solanaceae species. FT: Fleshy tomato; MT: Mushy tomato; CT: Cherry tomato; VBLE: Violet-black long eggplant; VBRE: Violet-black round eggplant; OLE: Ordinary long eggplant; LE: Linear eggplant; SP: Screw pepper; LP: Linear pepper; BP: Bell pepper; LGP: Long green pepper; PM: Pepino melon.

**Figure 4 insects-14-00591-f004:**
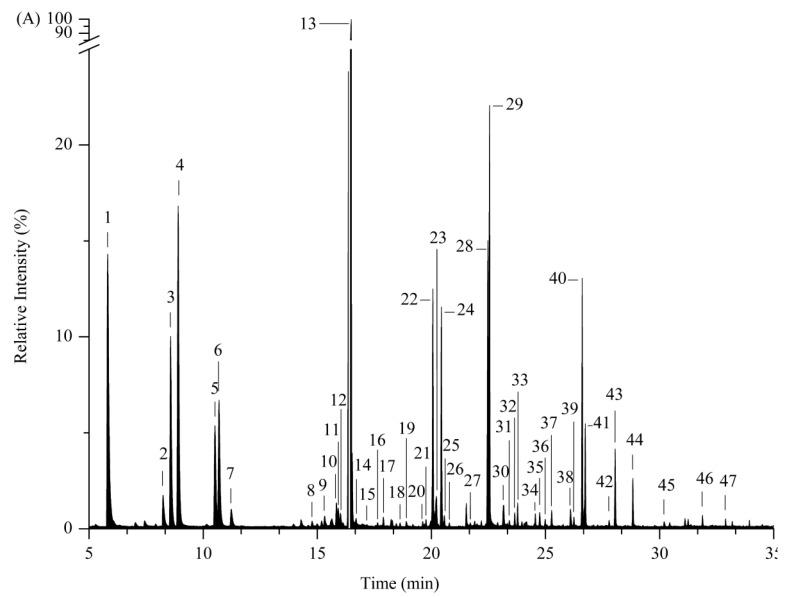
Peak chromatographic plot of the relative intensity of volatile compounds in *S. muricatum.* (**A**) The first biological replicates. (**B**) The second biological replicates. (**C**) The third biological replicates. The numbers from 1 to 47 on the peaks correspond to the numbers of the compounds in Table 2.

**Figure 5 insects-14-00591-f005:**
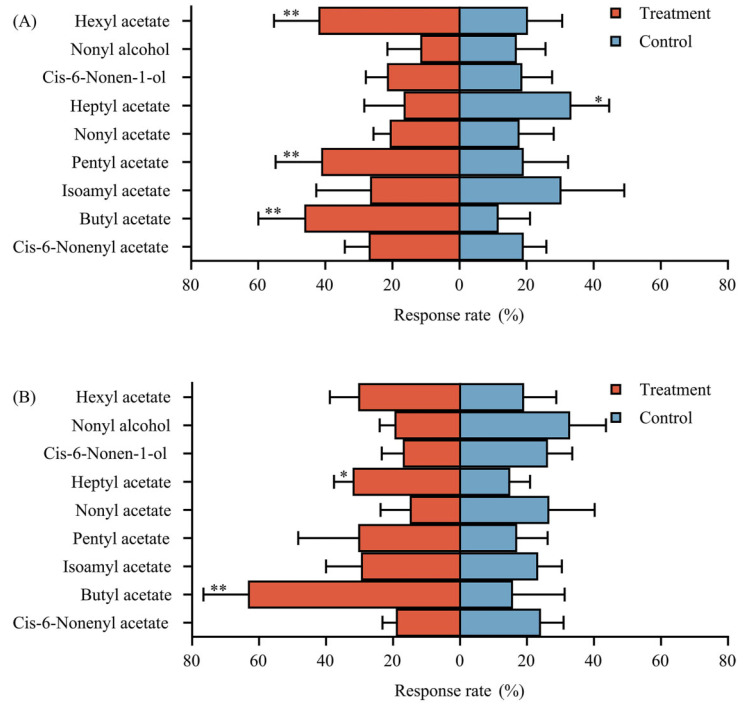
Olfactory behavioral responses of gravid *Z. tau* females to nine compounds from the *S. muricatum* volatile headspace. The dark orange bars show the percentages of the treatment group, blue bars show the percentages of the control group, asterisks (*) show that the percentages were significant at the *p* < 0.05 level, and binary asterisks (**) show that the percentages were significant at the *p* < 0.01 level. The total response rates were more than 65% for those marked with asterisks (*) and binary asterisks (**). (**A**) The response rate at a concentration of 5 mg/mL. (**B**) The response rate at a concentration of 0.5 mg/mL. (**C**) The response rate at a concentration of 0.05 mg/mL. (**D**) The response rate at a concentration of 0.005 mg/mL.

**Figure 6 insects-14-00591-f006:**
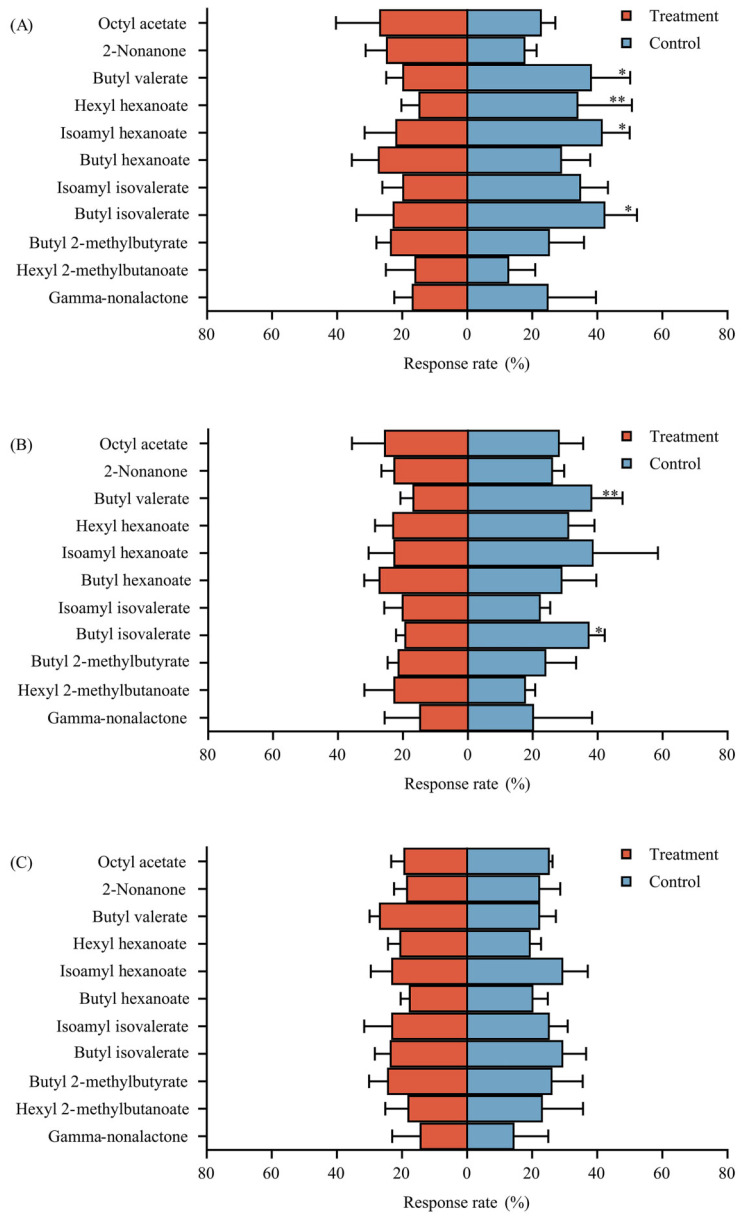
Olfactory behavioral responses of gravid *Z. tau* females to 11 compounds from the *S. muricatum* volatile headspace. The dark orange bars show the percentages for the treatment group, blue bars show the percentages for the control group, asterisks (*) show that the percentages were significant at the *p* < 0.05 level, and binary asterisks (**) show that the percentages were significant at the *p* < 0.01 level. The total response rates were more than 65% for those marked with asterisks (*) and binary asterisks (**). (**A**) The response rate at a concentration of 5 mg/mL. (**B**) The response rate at a concentration of 0.5 mg/mL. (**C**) The response rate at a concentration of 0.05 mg/mL. (**D**) The response rate at a concentration of 0.005 mg/mL.

**Table 1 insects-14-00591-t001:** Compounds used in olfaction behavior response experiment.

No.	CAS #	Compound	Purity	Supplier
1	123-86-4	Butyl acetate	99.7%	Aladdin^®^
2	123-92-2	Isoamyl acetate	99.5%	Aladdin^®^
3	628-63-7	Amyl acetate	99.5%	Aladdin^®^
4	142-92-7	Hexyl acetate	99.5%	Aladdin^®^
5	143-13-5	Nonyl acetate	99.7%	Aladdin^®^
6	112-06-1	Heptyl acetate	99.0%	Aladdin^®^
7	35854-86-5	Cis-6-nonen-1-ol	95.0%	Aladdin^®^
8	143-08-8	Nonyl alcohol	99.5%	Aladdin^®^
9	76238-22-7	Cis-6-nonenyl acetate	92.0%	Aladdin^®^
10	821-55-6	2-Nonanone	99.0%	Aladdin^®^
11	109-19-3	Butyl isovalerate	98.0%	Aladdin^®^
12	659-70-1	Isoamyl isovalerate	98.0%	Aladdin^®^
13	626-82-4	Butyl hexanoate	99.5%	Aladdin^®^
14	112-14-1	Octyl acetate	98.0%	Aladdin^®^
15	2198-61-0	Isoamyl hexanoate	98.0%	Aladdin^®^
16	15706-73-7	Butyl 2-methylbutyrate	98.0%	Aladdin^®^
17	6378-65-0	Hexyl hexanoate	98.0%	Macklin^®^
18	591-68-4	Butyl valerate	98.0%	Macklin^®^
19	10032-15-2	Hexyl 2-methylbutanoate	98.0%	Macklin^®^
20	104-61-0	Gamma-nonalactone	98.0%	Aladdin^®^
21	8042-47-5	Paraffin liquid	99.0%	Macklin^®^

CAS #: CAS (Chemical Abstracts Service) number, a unique numerical number for the identification of the compounds.

**Table 2 insects-14-00591-t002:** Volatile compounds and their relative contents in *S. muricatum*.

No.	Classification	CAS #	Compound	RI	Relative Content (%)
1	Esters	123-86-4	Butyl acetate	882	5.84 ± 0.96
2	Alcohols	111-27-3	1-Hexanol	957	0.46 ± 0.17
3	Esters	123-92-2	Isoamyl acetate	967	3.61 ± 0.32
4	Esters	5205-7-2	3-Methyl-3-buten-1-ol, acetate	978	6.14 ± 0.54
5	Esters	42125-10-0	(Z)-2-Penten-1-ol, acetate	1022	2.34 ± 0.14
6	Esters	628-63-7	Amyl acetate	1027	3.60 ± 0.61
7	Esters	1191-16-8	2-Buten-1-ol, 3-methyl-, acetate	1040	0.86 ± 0.46
8	Alcohols	3391-86-4	1-Octen-3-ol	1130	0.10 ± 0.01
9	Heterocycles	3777-69-3	2-Pentylfuran	1141	0.08 ± 0.01
10	Heterocycles	115051-66-6	S-(3-Hydroxypropyl) thioacetate	1158	0.30 ± 0.06
11	Esters	72237-36-6	4-Hexen-1-ol, acetate	1160	0.43 ± 0.05
12	Esters	3681-82-1	(E)-3-Hexen-1-ol, acetate	1162	0.20 ± 0.02
13	Esters	142-92-7	Hexyl acetate	1175	52.27 ± 1.09
14	Esters	56922-75-9	(Z)-2-Hexen-1-ol, acetate	1176	0.55 ± 0.08
15	Esters	15706-73-7	Butyl 2-methylbutanoate	1180	0.03 ± 0.01
16	Esters	109-19-3	Butyl isovalerate	1204	0.13 ± 0.05
17	Alkanes	1632-70-8	5-Methyl-undecane	1211	0.08 ± 0.04
18	Esters	30563-31-6	Butanoic acid, 4-pentenyl ester	1221	0.05 ± 0.01
19	Ketones	821-55-6	2-Nonanone	1231	0.10 ± 0.03
20	Esters	591-68-4	Butyl valerate	1258	0.07 ± 0.01
21	Esters	22597-23-5	4-Methylcyclohexanol acetate	1257	5.54 ± 2.72
22	Esters	659-70-1	Isoamyl isovalerate	1271	0.47 ± 0.03
23	Esters	112-6-1	Heptyl acetate	1275	2.91 ± 0.20
24	Esters	84254-81-9	Butanoic acid, 2-methyl-, 3-methyl-3-butenyl ester	1281	0.14 ± 0.03
25	Esters	54056-51-8	2-Butenoic acid, 3-methyl-, butyl ester	1285	0.05 ± 0.01
26	Alkanes	334-56-5	1-Fluoro-decane	1290	0.07 ± 0.04
27	Esters	84254-81-9	Butanoic acid, 2-methyl-, 3-methyl-3-butenyl ester	1317	0.06 ± 0.01
28	Alcohols	35854-86-5	Cis-6-Nonen-1-ol	1339	2.44 ± 1.12
29	Alcohols	143-8-8	Nonyl alcohol	1341	3.23 ± 1.55
30	Esters	626-82-4	Butyl hexanoate	1359	0.28 ± 0.04
31	Esters	3491-27-8	2,7-Octadien-1-ol, acetate	1367	0.07 ± 0.01
32	Esters	21722-83-8	Cyclohexaneethyl acetate	1374	0.14 ± 0.01
33	Esters	112-14-1	Octyl acetate	1378	0.26 ± 0.02
34	Esters	10032-15-2	Hexyl 2-methylbutanoate	1401	0.09 ± 0.02
35	Esters	142-9-6	2-Propenoic acid, 2-methyl-, hexyl ester	1407	0.15 ± 0.04
36	Esters	2198-61-0	Isoamyl hexanoate	1414	0.06 ± 0.01
37	Esters	30563-33-8	Hexanoic acid, 4-pentenyl ester	1423	0.15 ± 0.01
38	Esters	5413-59-2	Hexanoic acid, cyclopentyl ester	1449	0.17 ± 0.06
39	Ketones	112-12-9	2-Undecanone	1454	0.13 ± 0.03
40	Esters	76238-22-7	Cis-6-Nonenyl acetate	1465	2.79 ± 0.15
41	Esters	143-13-5	Nonyl acetate	1470	1.48 ± 0.21
42	Heterocycles	18679-18-0	(Z)-Dihydro-5-(2-octenyl)-2(3H)-furanone	1504	0.07 ± 0.01
43	Heterocycles	104-61-0	Gamma-nonalactone	1517	0.90 ± 0.16
44	Esters	6378-65-0	Hexyl hexanoate	1556	0.44 ± 0.04
45	Ketones	79-77-6	Beta-lonone	1744	0.06 ± 0.01
46	Alkylphenols	96-76-4	2,4-Di-tert-butylphenol	1821	0.09 ± 0.02
47	Alcohols	40716-66-3	Nerolidol	1975	0.09 ± 0.02

Data of the RI and relative content are presented as the mean ± SD. RI: the retention index. CAS #: CAS (Chemical Abstracts Service) number, a unique numerical number for the identification of the compound.

## Data Availability

The datasets in this study are available from the corresponding author upon reasonable request.

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
