# Peer review of "Evaluation of the Attractant Effect of Solanum muricatum (Solanales: Solanaceae) on Gravid Female Adults of Zeugodacus tau (Diptera: Tephritidae) and Screening of Attractant Volatiles"

_insects, 2023, doi:10.3390/insects14070591_

Round 1

Reviewer 1 Report

The manuscript titled “Evaluate of the attractant effect of Solanum muricatum (Aiton) to gravid female adults of Zeugodacus tau (Walker) and screening of attractant volatiles”, describes a research on Zeugodacus tau an important and serious pest in solanaceous species. The study seeks to know about the behavior of this diptera, specifically gravid females, and its interaction with a new potential host within Solanaceae (Solanum muricatum) through behavioral techniques used in the area of chemical ecology (olfactory bioassays, identification of compounds through GC-MS, among others). In itself, the research provides new information on the behavior of this pest and the use of chemical compounds that could potentially shed light on control strategies through monitoring. However, the manuscript presents a series of aspects that must be considered and clarified before a possible publication in the journal Insects.

Introduction:

Page 2, line 57: The authors provide information on the percentages of damage caused by the insect in other crops. Although S. muricatum will be used, is there any reference or information that indicates that if the pest attacks cucumbers, this could be of economic relevance?

Page 3, lines 99-108: The characteristics indicated for the volatiles seem to have a forage kairomone function. It is suggested to analyze if perhaps the authors are in search of this type of compound.

Material and Methods:

 Page 3, lines 120-133: Can individuals be separated by sex? If so, it is important that the authors indicate this, since if there is sexual differentiation between males and females, it would allow future experiments to be better directed.

 Page 4, 2.4 Oviposition bioassays.

- Why was Solanum muricatum used in subsequent bioassays?

- What does a mean optimal numbers of eggs laid? Authors should indicate a range of eggs to know what is optimal to expect.

- How many females were placed per treatment to achieve oviposition?

 Page 4, 2.5 Trapping bioassays.

- How were the 25-35d old gravid females determined?

 Page 5-6, 2.7 Olfactometric bioassays:

- Again, how do the authors know which females are gravid?

-Why aren't the volatiles of the other hosts evaluated? So why was the oviposition test carried out with all the other hosts? Or is something known about the chemical profile of the other hosts and have they been evaluated before?

Results:

Page 6,

Line 280: change “expriments” by “experiments”

Line 283: 6.07 grains/female… what does grains mean in this evaluation?

Figures 3 and 4: The size and image quality of the indicated figures should be improved.

What is the difference (besides the "n") of evaluating 9 and 11 compounds? What was the criteria for separating these figures?

Discussion:

Page 12, lines 418-420: It is not clear if these parameters were evaluated or if they are only part of the discussion that is beginning to focus on other insects and not on Z. tau.

Page 12, line 454: change “solanaceae” by “Solanaceae”

Page 13, lines 496-498: It should be explained what is the difference with these previous studies referring to the content of esters.

Page 14, line 529: Don't understand the concept of “tendency behavior”. Please explain what the authors mean.

Page 14, line 543: What do you mean by systematic studies? what would they be specifically?

Page 14, lines 563-574: At the end of the discussion, the authors propose the use of plant traps.....why is the use of traps with attractive or repellent compounds discarded or not an option? In addition, it was carried out in studies of volatile compounds, which is the basis for this type of trapping. This is named as an option in the last paragraph of the conclusions.

Reviewer 2 Report

The authors screened a range of Solanaceae crop species to determine whether there was an attractive role for volatiles towards the insect pest Zeugodacus tau. Volatile components were identified and assayed, and several compounds demonstrated significant attraction towards the insect. 

Certain parts of the methods are not very clearly described, making it difficult to interpret the results, although this may be due to the complexity of the bioassays. I think in some instances, some diagrams highlighting the study design would be beneficial to the reader: e.g. section 2.4 and 2.5.

GC-MS analyses would be improved by calculating kovats retention indices for tentatively identified compounds, by injecting a series of alkanes into the GC-MS (C7-C22 alkanes).

Figures 3-4 are not very clear, and the compound names cannot be read on the y axis, which make it very difficult to interpret the results

Other comments are listed below:

Simple summary

line 14- were found to oviposit

line 17- GC-MS was used to identify

Lin 18-Y tube olfactometer system was used

Line 19- the results show

Abstract

Line 25- is a pest seriously harmful to Solanaceae...

Line 26- oviposit instead of ovipositing

Line 30- inducing behavioural responses

Line 31- 'results' instead of 'result'

Line 35- Hexyl hexanoate instead of tHexyl hexanoate

Line 40- revise the concluding sentence

Introduction

I think the authors have provided a nice overview of the current literature. Most suggestions are minor, grammatical/vocabulary related

Line 50- now it is one of the most economically important agricultural pests...

Line 72- It is also low in calories

Line 76- in the past several decades

Line 78- has instead of had 

Line 82- 'found to oviposit' instead of 'found to ovipositing'

Line 92- 's' should be lowercase in 'screening'

Line 98- 'using GC-MS' instead of 'while used GC-MS'

Line 100- remove 'ingredients'

Line 101- what is Raspberry Ketone?

Line 104- revise final sentence

Materials and methods

Line 164- cages instead of cage

Line 179- an image of the traps would be useful here- maybe in the supplementary materials?

Line 180- cages instead of cage

Line 187- Traps that contain instead of traps that containing

Line 189- remove 'in the freezer'

Line 199- change to 'Headspace volatile samples of pepino melon fruits were identified and quantified using Gas Chromatography-Mass Spectrometry'

Line 204- Three biological replicates were performed per sample' instead of 'The sample were repeated three times'

Results

Line 279- opening sentence needs revision

Line 315- opening sentence needs revision

Line 362- were found to elicit

Line 370- 'significantly repellent' instead of 'significantly repellency'

Line 371- found to repel instead of 'found that repelled to'

Line 373- revise final sentence

Line 360- tested instead of tasted

Line 380- figures very unclear

Line 381- Change to 'Olfactory behavioural responses of gravid Z. tau females to nine compounds from the S. muricatum volatile headspace'

Discussion

Line 410- 'had a significant impact' instead of 'had a significantly impact'

Line 412- remove 'that'

Line 417- hosts with greater attraction were advantageous to offspring performance of fruit flies

Line 426- remove 'that'

Line 428- change 'in our results showed' to 'here, we show that line pepper...'

Line 432- Furthermore, in contrast...

Line 439- 'in a previous study' instead of 'in previous study'

Line 444- revise opening sentence

Line 465- cue instead of cues

Line 466- remove 'what'

Line 468- 'The olfaction of fruit flies are important contributors to host location...'

Line 471- revise

Line 488- 'The combination of GC-MS with behavioural bioassays...'

Line 489- remove 'while'

Line 490- 'on insect behaviour'

Line 491- volatile odor compounds including ketones...

Line 494- 'our results show that 47 volatiles were identified'

Line 506- 'Host plants conducive to larval survival and development are selected by gravid females...'

Line 546- 'Pepino melon has the potential to act as a trap plant...'

Line 547- 'Strong attraction of plants...'

Conclusions

Line 576- 'had a significant impact' instead of 'had a significantly impact'

Line 580- Our results showed that the odour of Solanaceae crops is a key cue affecting the ability to attract gravid Z. tau females

There were quite a few vocabulary/grammatical errors throughout the manuscript

Reviewer 3 Report

comments and suggestions can be found in the revised manuscript

Round 2

Reviewer 1 Report

The authors have responded to each of the suggestions made and have incorporated these changes into the manuscript. Therefore, I suggest its publication in the Insects journal.

Author Response

Point 1: Comments and Suggestions for Authors

The authors have responded to each of the suggestions made and have incorporated these changes into the manuscript. Therefore, I suggest its publication in the Insects journal.

Response 1: We appreciated very much for your constructive and insightful comments. Once again, thank you very much for your comments.

Reviewer 2 Report

The authors have made substantial improvements to the manuscript, which has improved the overall quality of the work. There are a few further points that should be addressed :

Figures 4/5/6 are still not of sufficient quality for publication- it is difficult to read the peak numbers/compound names. Could these figures be increased in size/resolution?

In the results section, it is difficult to determine what the results in 3.1 and 3.2 are showing, specifically for the analyses of variance. I've made some suggestions as to how these could be reported, following your statistical analyses, but the authors should check that my understanding is correct.

Simple summary

Line 12- Combine the first two sentences- The Solanaceae crops are the main hosts of Zeugodacus tau; a pest insect which causes serious damage to economically important crops of Solanaceae

Line 15- However, the differences in the ability of the fruits...

Line 19- Remove the sentence 'The results show that S. muricatum had the optimal ability to attract gravid Z. tau females.', and amend the following sentence to 'The results show that S. muricatum odour was attractive towards Z. tau females.'

Introduction

Line 64- I would remove 'Meanwhile, the tolerance of Z. tau to extreme cold and heat stress is strong'. This is quite subjective, and the following sentence clarifies this point.

Line 68- remove 'tremendously'

Line 71- replace 'about' with 'approximately'

Line 83- 'distributed' instead of 'distribution'

Line 88- 'int Fuzhou', should this be 'in the Fuzhou'?

Methods

Line 138- how was the pumpkin juice obtained- commercially available? Made in the lab? More information needed here I think, as it is part of the insect culturing.

Line 146- yeast extract, same as with the pumpkin juice, more information needed.

Line 155- remove 'for use'

Line 241- 'Tentatively identified'

Results

3.1- I'm not sure what the statistical test is showing here- do you mean that the numbers of eggs laid by gravid females for each plant was significantly different across the different cultivars? If so, this needs to be reworded. 'The numbers of eggs laid by gravid female Z. tau females was significantly different across four cultivars of eggplants', and 'the highest number of eggs laid (6.07 eggs/female) was on the VBRE cultivar'. This should be corrected throughout this section, and 3.2

Line 315- 'were significantly different' instead of 'significantly difference'

Line 375- 'tentatively identified'

Section 3.3- thank you for including the kovats retention indices as suggested, and I would strongly encourage including retention indices wherever a compound has been tentatively identified, for all future manuscripts. I do not think it is necessary to include the plus/minus values in this instance- instead, you can round the calculated value up/down to a whole number.

Figure 4- Can't read the peak numbers

Figure 5 and 6 are also not very clear, I think the figure sizes need to be bigger

Discussion

Line 476- 'easy' instead of 'easily'

Line 491- 'India' instead of 'indian'

line 524- i'm not sure whether this line is necessary, as these chemical classes cover a very broad range of compounds. I would suggest removing this sentence

Line 573- You could also include a study which showed that blends of compounds can be attractive towards insects, although when compounds are tested individually, the compounds may actually act as repellents- see https://doi.org/10.1016/j.anbehav.2009.11.028

Line 579- I would also mention that electrophysiology studies (e.g. GC-EAG) can help to locate specific volatiles in a mixture which the insects antennae are responding to, which can help to pinpoint which compounds specifically may be causing biological activity. This could also be some follow on work, where blends of the attractive compounds could be tested for attraction towards Z. tau

line 586- 'are' instead of 'sre'

Very good, only a few minor typographical issues I spotted which are included in my comments

Reviewer 3 Report

authors complied with the request

Author Response

Point 1:

Comments and Suggestions for Authors: authors complied with the request.

Response 1:

We appreciated very much for your constructive and insightful comments. Once again, thank you very much for your comments.

Round 3

Reviewer 2 Report

I am satisfied that the authors have made all the changes I have suggested, and I would like to congratulate them on this body of work. I have only a few very minor changes to suggest.

Line 30- I would edit this line, and change it to 'The results show that S. muricatum odours have an attractive role towards gravid Z. tau females'.

Line 38- 'required' instead of 'requires'

Author Response

We appreciated very much for your constructive and insightful comments.

Point 1: Line 30- I would edit this line, and change it to 'The results show that S. muricatum odours have an attractive role towards gravid Z. tau females'.

Response 1: Thanks for your important comment. We have revised this question.

Point 2: Line 38- 'required' instead of 'requires'

Response 2: Thanks for your important comment. We have revised this question.

Once again, thank you very much for your comments.